# A Polynomial Time Graph Isomorphism Algorithm via Self-Supervised Gradient Descent

## Abstract

Graph isomorphism (GI) is a fundamental problem in graph theory. Despite recent advancements, determining whether two graphs are isomorphic remains computationally challenging. This paper introduces the Polynomial Time Graph Isomorphism (PTGI) algorithm, an optimization-based approach leveraging self-supervision techniques to efficiently tackle the graph isomorphism problem. PTGI aims to escape local optima caused by graph symmetries and provides high accuracy in identifying isomorphic graphs in polynomial time. Experimental results demonstrate PTGI's effectiveness across various graph types, making it a valuable tool for practical applications.

## 1 Introduction

Graph isomorphism (GI) is a fundamental problem in graph theory that deals with determining whether two graphs are structurally identical, namely "isomorphic", up to a relabeling of vertices. Formally, two graphs $G_A = (V_A, E_A)$ and $G_B = (V_B, E_B)$ are said to be *isomorphic* if and only if there exists a bijection $f : V_A \to V_B$ such that for any pair of vertices $u, v \in V_A$, $(u, v) \in E_A$ if and only if $(f(u), f(v)) \in E_B$. In other words, the two graphs have the same connectivity pattern, but the vertices may be labeled differently. Graph isomorphism has numerous applications in various fields, including chemistry Balaban (1985); Merkys et al. (2023), biomedical sciences Sporns et al. (2005); Singh et al. (2007), network analysis Cook & Holder (2006), computer vision Christmas et al. (1995); Zaslavskiy et al. (2008), and pattern recognition Pelillo et al. (1999).

Determining whether two graphs are isomorphic is computationally challenging. It belongs to the class of NP (nondeterministic polynomial time) problems and is one of the few remaining problems in NP that is not known to be either P (polynomial time) or NP-complete Fortin (1996). The main areas of research for graph isomorphism problem are design of fast algorithms and theoretical investigations of its computational complexity, both for the general problem and for special classes of graphs. While recent breakthroughs have shown that GI is solvable in quasipolynomial $(\exp((\log n)^{O(1)}))$ time Babai (2016), the demand for polynomial time algorithms remains, especially for real-world graphs that can be super large. Many existing polynomial time graph isomorphism algorithms only work with specific classes of graphs Hopcroft & Tarjan (1972); Babai et al. (1980); Luks (1982); Grohe & Marx (2012); Babai et al. (2013); Lokshtanov et al. (2017), bringing challenges for generalization to real-world graphs.

Due to the computational complexity of exact isomorphism checking, optimization-based graph isomorphism algorithms have been proposed to efficiently tackle the graph isomorphism problem Umeyama (1988); Zaslavskiy et al. (2008); Vogelstein et al. (2011); Aflalo et al. (2015). While these algorithms offer improved efficiency compared to exact methods, they typically provide approximate solutions rather than exact isomorphism checking. By formulating the problem as an optimization task and iteratively refining the solution, optimization-based approaches strive to find a mapping between the vertices of two graphs that maximizes a similarity metric or minimizes a dissimilarity metric. Despite not guaranteeing exact isomorphism, optimization-based algorithms are valuable tools for practical applications where efficiency is prioritized over correctness.

While optimization-based graph isomorphism algorithms have demonstrated efficient graph isomorphism checking, we found that in practice they often result in local optima when graphs possess symmetries, even for very simple graphs. In this case, an optimization-based GI algorithm fails

identifying isomorphic graphs. A recent study Klus & Gelß (2023) highlighted that graph symmetries can lead to repeated eigenvalues that complicates graph isomorphism testing. This is a significant limitation of existing studies.

To address this limitation, in this paper, we introduce the *Polynomial Time Graph Isomorphism (PTGI)* algorithm, an optimization-based approximate graph isomorphism algorithm with a worst-case polynomial time complexity of $O(n^4)$. PTGI incorporates self-supervision techniques to avoid local optima and demonstrates high accuracy in identifying isomorphic graphs. Experimental results showcase PTGI's effectiveness in polynomial time, without constraints on graph types or properties.

The contributions of this paper are summarized as follows:

1. We demonstrate the effectiveness of self-supervision in escaping local optima caused by graph symmetries in optimization-based graph isomorphism algorithms.

2. We propose the PTGI algorithm, an approximate graph isomorphic algorithm with a worst-case polynomial time complexity, leveraging self-supervision to avoid local optima.

3. Experimental results validate PTGI's ability to accurately identify isomorphic graphs in polynomial time, making it a near-exact graph isomorphism algorithm applicable to various graph types.

The following content of this paper is organized as follows: In Sec. 2, we introduce some graph isomorphism algorithms that are closely-related to our proposed algorithm. In Sec. 3, we formulate the popular paradigm for designing optimization-based graph isomorphism algorithms, and demonstrate that this paradigm often result in local optima. In Sec. 4, we formally propose the Polynomial Time Graph Isomorphism (PTGI) algorithm, and analyze its time and space complexity. In Sec. 5, we present and discuss experimental results for evaluating both the effectiveness and efficiency of our proposed algorithm. In Sec. 6, we conclude our study. In Sec. 7, we propose several potential future research directions based on our current work.

## 2 RELATED WORKS

Graph Isomorphism Algorithms can vary significantly in terms of complexity, efficiency, and applicability. Some algorithms focus on exact isomorphism checks and are suitable for small to medium-sized graphs, while others employ heuristic or approximation techniques to handle larger graphs more efficiently. Prominent exact graph isomorphism algorithms include:

- Nauty McKay (2007): A widely-used algorithm developed by Brendan McKay for exact graph isomorphism testing and graph canonization. The worst-case time complexity of Nauty is exponential ($O(2^n)$).

- VF2 Cordella et al. (2004): A backtracking-based algorithm proposed by Cordella et al. which efficiently explores possible mappings between vertices using constraints. VF2 has a worse-case ($O(n!)$) factorial time complexity.

- Ullmann's algorithm Ullmann (1976): It uses backtracking combined with constraint propagation to explore the space of possible mappings between vertices of the two graphs. Ullmann's algorithm has a worse-case ($O(n!)$) factorial time complexity.

Although these exact graph isomorphism algorithms have demonstrated efficient graph isomorphism checking for many examples of large graphs, their theoretical worst-case complexity remains computationally intractable.

Due to the computational complexity of exact isomorphism checking, optimization-based graph isomorphism algorithms leverage optimization techniques to efficiently tackle the graph isomorphism problem. A commonly used optimization objective is to find a bijective vertex mapping, represented by a permutation matrix, such that the adjacency disagreement of the two graphs being mapped is minimized Aflalo et al. (2015). A variaty of existing approaches avoid the combinatory complexity when searching for permutation matrices by relaxing the domain of permutation matrices to a trackable convex or near-convex space when optimizing graph adjacency disagreement (denoted in (2)). A popular approach is to relax the space of permutation matrices to the space of doubly stochastic matrices then solve the optimization problem in polynomial time using quadratic programming or

gradient descent Umeyama (1988); Zaslavskiy et al. (2008); Vogelstein et al. (2011); Aflalo et al. (2015); Fiori & Sapiro (2015). An alternative relaxation technique is to replace permutation matrices with orthogonal matrices Zavlanos & Pappas (2008); Klus & Sahai (2018). In practice, we found that such type of optimization approaches often generate local optima for graphs possessing symmetries. Consequently, their accuracy on identifying isomorphic graph pairs is not guaranteed. For example, Aflalo showed that such type of optimization approach only has high accuracy on "friendly graphs", namely graphs whose adjacency matrices have simple spectrum (i.e., all of its eigen values are distinct) Aflalo et al. (2015) . Umeyama's algorithm Umeyama (1988) requires graphs being matched to be sufficiently close to each other in terms of eigenvectors. This limitation is significant from the perspective of generalization since real-world graphs are not neccessarily "friendly".

## 3 GRAPH ISOMORPHISM OPTIMIZATION PARADIGM

### 3.1 THE GRAPH ISOMORPHISM PROBLEM

A graph is defined as $G = (V, E)$ where $V$ is a set of vertices and $E$ is a set of edges. In the context of this paper, we consider graphs as non-weighted and non-labeled. Two graphs $G_A = (V_A, E_A)$ and $G_B = (V_B, E_B)$ are said to be *isomorphic*, denoted as $G_A \simeq G_B$, if and only if there exists a bijection $f : V_A \rightarrow V_B$ such that for any pair of vertices $u, v \in V_A$, $(u, v) \in E_A$ if and only if $(f(u), f(v)) \in E_B$. Such a bijection is called an *isomorphism* of graph $G_A$ and $G_B$.

Graph isomorphism can also be equivalently defined using adjacency matrices. Given a graph $G = (V, E)$ with $|V| = n$, let $\mathbf{A}$ be the adjacency matrix of $G$, which is a $n \times n$ binary matrix defined as follows:

$$\mathbf{A} = [a_{ij}] \begin{cases} 1 & \text{if } (v_i, v_j) \in E, \\ 0 & \text{if } (v_i, v_j) \notin E. \end{cases} \tag{1}$$

Given two graphs $G_A = (V_A, E_A)$ and $G_B = (V_B, E_B)$ ($|V_A| = |V_B| = n$) with adjacency matrices $\mathbf{A}$ and $\mathbf{B}$, $G_A$ and $G_B$ are isomorphic if and only if there exists a permutation matrix $\mathbf{P}$ such that $\mathbf{A} = \mathbf{PBP}^\mathsf{T}$. A permutation matrix is a square binary matrix that has exactly one entry of 1 in each row and each column with all other entries 0. Each permutation matrix $\mathbf{P}$ corresponds to a bijection $f : V_A \rightarrow V_B$.

The computationional problem of determining whether two finite graphs are isomorphic is called the *Graph Isomorphism (GI) Problem*. A graph isomorphism algorithm is a computational method used to determine whether two given graphs are isomorphic.

### 3.2 A PARADIGM FOR OPTIMIZATION-BASED GRAPH ISOMORPHISM ALGORITHMS

Due to the computational complexity of exact isomorphism checking, optimization-based graph isomorphism algorithms utilize optimization techniques to tackle the graph isomorphism problem more efficiently but provide approximate (near-exact) rather than exact isomorphism checking. An optimization-based graph isomorphism algorithm typically aims to find mapping between the vertices of two graphs that maximizes a similarity metric or minimizes a dissimilarity metric.

A commonly used dissimilarity metric in optimization is the adjacency (or connectivity) disagreement. For two graphs $G_A$ and $G_B$, with with adjacency matrices $\mathbf{A}$ and $\mathbf{B}$, and a permutation matrix $\mathbf{P}$, the adjacency disagreement between $G_A$ and $G_B$ with respect to $\mathbf{P}$ is defined as $||\mathbf{A} - \mathbf{PBP}^\mathsf{T}||^2$, where $|| \cdot ||$ denotes a norms such as the Euclidean norm (used in this paper). Let $\mathcal{P}(n)$ denote the set of $n \times n$ permutation matrices $\mathcal{P}(n) = \{\mathbf{P} \in \{0, 1\}^{n \times n} : \mathbf{P1} = \mathbf{P}^\mathsf{T}\mathbf{1} = \mathbf{1}\}$, where $\mathbf{1}$ is an $n$-dimensional column vector. Then, the graph isomorphism can be formulated as the following optimization problem:

$$\mathbf{P}^* = \underset{\mathbf{P} \in \mathcal{P}(n)}{\arg \min} ||\mathbf{A} - \mathbf{PBP}^\mathsf{T}||^2. \tag{2}$$

The two graphs $G_A$ and $G_B$ are isomorphic if and only if $||\mathbf{A} - \mathbf{P}^*\mathbf{BP}^{*\mathsf{T}}||^2 = 0$.

Equation (2) is a variant of the Quadratic Assignment Problem (QAP), which is NP-hard and therefore has no known polynomial time solution yet. This complexity is due to the combinatorial complexity of the constraint $\mathbf{P} \in \mathcal{P}(n)$. Relaxation techniques can reduce this complexity by replacing

the domain of $\mathbf{P}$ with a convex continuous set. A popular approach is to relax the space of $\mathcal{P}(n)$ to its convex hull, i.e., the space of doubly stochastic matrices $\mathcal{D}(n) = \{\mathbf{P} : \mathbf{P1} = \mathbf{P^\intercal 1} = \mathbf{1}, \mathbf{P} \succeq 0\}$, where $\mathbf{1}$ is an $n$-dimensional column vector and $\succeq$ indicates an element-wise inequality. Then, the convex relaxed graph isomorphism can be formulated as the following optimization problem:

$$\mathbf{P}^* = \arg\min_{\mathbf{P} \in \mathcal{D}(n)} ||\mathbf{A} - \mathbf{PBP^\intercal}||^2. \tag{3}$$

A regularization term $||\mathbf{PP^\intercal} - \mathbf{I}||^2$ can be added to enforce $\mathbf{P}^*$ to be close to a real permutation matrix. The convex relaxed graph isomorphism with regularization is formulated as:

$$\mathbf{P}^* = \arg\min_{\mathbf{P} \in \mathcal{D}(n)} (||\mathbf{A} - \mathbf{PBP^\intercal}||^2 + \alpha ||\mathbf{PP^\intercal} - \mathbf{I}||^2). \tag{4}$$

A local minimum of both (3) and (4) can be efficiently found via gradient descent. Then the doubly stochastic matrix $\mathbf{P}^* \in \mathcal{D}(n)$ resulted from either (3) or (4) can be projected to a permutation matrix $\hat{\mathbf{P}} \in \mathcal{P}(n)$ by

$$\hat{\mathbf{P}} = \arg\min_{\mathbf{P} \in \mathcal{P}(n)} -\langle \mathbf{P}, \mathbf{P}^* \rangle, \tag{5}$$

where $\langle \cdot, \cdot \rangle$ is the Euclidean inner product. Equation (5) can be solved as a Linear Assignment Problem (LAP) efficiently in polynimial time via the Hungarian AlgorithmKuhn (1955). Then, the two graphs $G_A$ and $G_B$ are isomorphic if and only if $\mathbf{A} = \hat{\mathbf{P}} \mathbf{B} \hat{\mathbf{P}}^\intercal$. The above mentioned steps form a popular paradigm for a number of existing optimization-based GI algorithms Vogelstein et al. (2011).

### 3.3 Limitation of the Paradigm

In practice, we found that optimizing (3), (4) or their variants via gradient descent often lead to local optima when graphs possess symmetries, even for very simple graphs. In this case, the optimization-based GI algorithm fails identifying isomorphic graphs. This is a significant limitation of existing approaches. For instance, consider the two isomorphic graphs $G$ and $H$ with four vertices and two edges each shown in Fig. 1. There exists multiple permutation matrices $\mathbf{P} \in \mathcal{P}(4)$ such that

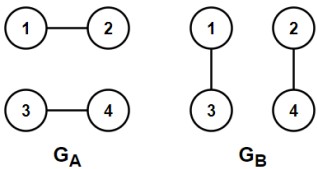

Figure 1: Two graphs with four vertices each.

$\mathbf{A} = \mathbf{PBP^\intercal}$, such as:

$$\mathbf{P}_1 = \begin{bmatrix} 1 & 0 & 0 & 0 \\ 0 & 0 & 1 & 0 \\ 0 & 1 & 0 & 0 \\ 0 & 0 & 0 & 1 \end{bmatrix}, \mathbf{P}_2 = \begin{bmatrix} 0 & 1 & 0 & 0 \\ 0 & 0 & 0 & 1 \\ 1 & 0 & 0 & 0 \\ 0 & 0 & 1 & 0 \end{bmatrix}$$

They are both global optima, i.e., isomorphisim of the two graphs. However, optimizing either (3) or (4) via gradient descent may result in an undesired local optima $\mathbf{P}^*$ as follows:

$$\mathbf{P}^* = \begin{bmatrix} 0.25 & 0.25 & 0.25 & 0.25 \\ 0.25 & 0.25 & 0.25 & 0.25 \\ 0.25 & 0.25 & 0.25 & 0.25 \\ 0.25 & 0.25 & 0.25 & 0.25 \end{bmatrix}$$

, especially when a flat doubly stochastic matrix, $\mathbf{P}^{(0)} = \mathbf{1} \cdot \mathbf{1}^\intercal / n$ is used as an initial position of gradient descent. There is no straightforward approach to convert the above $\mathbf{P}^*$ to a permutation matrix which corresponds to an isomorphism such as $\mathbf{P}_1$ or $\mathbf{P}_2$ rather than some incorrect permutation matrix such as:

$$\mathbf{P}_3 = \begin{bmatrix} 1 & 0 & 0 & 0 \\ 0 & 1 & 0 & 0 \\ 0 & 0 & 1 & 0 \\ 0 & 0 & 0 & 1 \end{bmatrix}$$

, which can be produced by commonly-used projection techniques such as the Hungarian algorithm.

It is worth to mention that with randomly initialized starting point, gradient descent can still yield local optima Du et al. (2017). In practice, we also found that optimizing either (3) or (4) with randomly initialized parameters can also lead to local optima when graphs possess symmetries, even for very simple graphs. Perturbed Gradient DescentJin et al. (2017) was proposed to escape local optima by adding random noise into model parameters. For the graph isomorphism problem, we found that local optima can be escaped by involving self-supervision into the gradient descent process.

For instance, given the local optima $\mathbf{P}^*$ for Fig. 1, we can see that vertex $v_1 \in V_A$ has an equal probability to correspond to $u_1, u_2, u_3, u_4 \in V_B$. This is heuristically true because $v_1$ is mapped to each $u \in V_B$ once among all the isomorphisms between the two graphs. Therefore, without any prior preference, each vertex $v \in V_A$ is equally likely to correspond to each vertex $u \in V_B$ in a randomly chosen isomorphism. However, if we arbitrarily let $v_1$ correspond to $u_1$, $v_3$ correspond to $u_2$, then (3) or (4) forces $v_2$ correspond to $u_3$, and $v_4$ correspond to $u_4$. It is worth to mention that this type of arbitrary vertex correspondence selection does not require human labeling but can be performed in an self-supervised way based on some heuristic rules. This example inspires us that by involving self-supervision into the gradient descent process for graph isomorphism checking, local optima might be escaped. Based on this inspiration, we designed a novel *Polynomial Time Graph Isomorphism (PTGI)* algorithm which is an optimization-based approximate graph isomorphic algorithm via self-supervised gradient descent.

## 4 PROPOSED METHOD

In this section, we first formulate the proposed *Polynomial Time Graph Isomorphism (PTGI)* algorithm as follows:

---

**Algorithm 1** Polynomial Time Graph Isomorphism (PTGI)

---

**Require:** Number of vertices $n$, adjacency matrices $\mathbf{A}, \mathbf{B}$, gradient descent maximum steps $T$, learning rate $\eta$, self-supervision weight $\alpha$
**Ensure:** A graph isomorphic indicator $\in \{True, False\}$
1: $\mathbf{S} \leftarrow \{0\}^{n \times n}$
2: $\mathbf{L} \leftarrow \{0\}^{n \times n}$
3: **for** $r = 0$ **to** $n$ **do**
4:    **for** $t = 1$ **to** $T$ **do**
5:       $\mathbf{P} \leftarrow \text{Softmax}(\mathbf{S})$
6:       $\mathcal{L}(\mathbf{S}) = ||\mathbf{A} - \mathbf{P}\mathbf{B}\mathbf{P}^T||^2 + \alpha| - \mathbf{L} \odot \log \mathbf{P}|$
7:       $\mathbf{S} \leftarrow \mathbf{S} - \eta \nabla \mathcal{L}(\mathbf{S})$
8:    **end for**
9:    $\mathbf{\Pi} \leftarrow \text{Onehot}(\mathbf{P})$
10:   **if** $\mathbf{A} = \mathbf{\Pi}\mathbf{B}\mathbf{\Pi}^\intercal$ and $\mathbf{\Pi}\mathbf{\Pi}^\intercal = \mathbf{I}$ **then**
11:      Return $True$
12:   **end if**
13:   $i, j = \arg\max_{i,j} p_{ij}$, s.t., $\sum_{k=1}^{n} l_{ik} = 0$
14:   $l_{ij} = 1$
15: **end for**
16: Return $False$

---

Next, we explain the details of the above algorithm. Firstly, instead of searching for an optimal doubly stochastic matrix, our PTGI algorithm searches for an optimal vertex similarity matrix $\mathbf{S} \in \mathbb{R}^{n \times n}$. This makes the optimization problem convex. $\mathbf{S}$ is initialized as an all-zero matrix (line 1).

In line 2, we initialize a label matrix $\mathbf{L} \in \{0, 1\}^{n \times n}$ with all zeros. The label matrix $\mathbf{L}$ serves as a self-supervision signal. Each entry $l_{ij}$ means that the $i$-th vertex in graph $G_A$ should correspond to the $j$-th vertex in graph $G_B$.

In each iteration of the gradient descent (line 4-8), we first compute the stochastic vertex mapping matrix $\mathbf{P}$ as a standard softmax of $\mathbf{S}$, namely

$$p_{ij} = \frac{\exp(s_{ij})}{\sum_{k=1}^{n} \exp(s_{ik})}. \tag{6}$$

Then in line 6, we define a loss function $\mathcal{L}(\mathbf{S})$ as the sum of the adjacency disagreement $||\mathbf{A} - \mathbf{P}\mathbf{B}\mathbf{P}^T||^2$ and a self-supervision loss $|-\mathbf{L} \odot \log \mathbf{P}|$ multiplied by a self-supervision weight $\alpha$, where $||\cdot||, |\cdot|$ denote the $l$-1 and $l$-2 norm. The self-supervision loss is the standard cross-entropy loss that is used to enforce $\mathbf{P}$ close to the pseudo ground-truth label $\mathbf{L}$, namely

$$|-\mathbf{L} \odot \log \mathbf{P}| = -\sum_{i=1}^{n} \sum_{j=1}^{n} l_{ij} \log(p_{ij}). \tag{7}$$

Note that if a row of $\mathbf{L}$ has all zeros, it means no self-supervision signal is provided for this row. In this case, the cross-entropy loss for this particular row is zero, thus not need to be optimized.

After we get a local optimal $\mathbf{P}$ via one iteration of gradient descent, we project it to a permutation matrix $\mathbf{\Pi}$ via standard one-hot encoding (line 9), namely

$$\pi_{ij} = \begin{cases} 1 & \text{if } j = \arg\max_k p_{ik}, \\ 0 & \text{if otherwise.} \end{cases} \tag{8}$$

Then, we test if $\mathbf{\Pi}$ is a correct graph isomorphism, and if so, the PTGI algorithm returns a True indicating the two graphs are isomorphic (line 10-12).

Next, we discuss how to iteratively update the self-supervision signal $\mathbf{L}$ (line 13-14). After the $r$-th iteration ($0 \leq r \leq n$) of gradient descent, we select one vertex correspondence pair $(i, j)$ with the highest probability score in the stochastic vertex mapping matrix $\mathbf{P}$ as the self-supervision signal. We also ensure $(i, j)$ has not been selected in previous iterations by adding a constraint $\sum_{k=1}^{n} l_{ik} = 0$. Then we add a new self-supervision signal $l_{ij} = 1$.

The self-supervision signal incrementally builds a graph isomorphism by adding one vertex correspondence in each iteration. In the $r$-th iteration, there exist $r$ vertex correspondence pairs as self-supervision signals. The self-supervision signal serves as a tie-breaker to escape the local optima.

We illustrate how $\mathbf{L}$ and $\mathbf{P}$ are updated using the example in Fig. (1).

Iteration 0:

$$\mathbf{L}^{(0)} = \begin{bmatrix} 0 & 0 & 0 & 0 \\ 0 & 0 & 0 & 0 \\ 0 & 0 & 0 & 0 \\ 0 & 0 & 0 & 0 \end{bmatrix}, \mathbf{P}^{(0)} = \begin{bmatrix} \mathbf{0.25} & 0.25 & 0.25 & 0.25 \\ 0.25 & 0.25 & 0.25 & 0.25 \\ 0.25 & 0.25 & 0.25 & 0.25 \\ 0.25 & 0.25 & 0.25 & 0.25 \end{bmatrix}$$

Iteration 1:

$$\mathbf{L}^{(1)} = \begin{bmatrix} \mathbf{1} & 0 & 0 & 0 \\ 0 & 0 & 0 & 0 \\ 0 & 0 & 0 & 0 \\ 0 & 0 & 0 & 0 \end{bmatrix}, \mathbf{P}^{(1)} = \begin{bmatrix} 1 & 0 & 0 & 0 \\ 0 & 0 & 1 & 0 \\ 0 & 0.5 & 0 & 0.5 \\ 0 & 0.5 & 0 & 0.5 \end{bmatrix}$$

Iteration 2:

$$\mathbf{L}^{(2)} = \begin{bmatrix} 1 & 0 & 0 & 0 \\ 0 & 0 & \mathbf{1} & 0 \\ 0 & 0 & 0 & 0 \\ 0 & 0 & 0 & 0 \end{bmatrix}, \mathbf{P}^{(2)} = \begin{bmatrix} 1 & 0 & 0 & 0 \\ 0 & 0 & 1 & 0 \\ 0 & \mathbf{0.5} & 0 & 0.5 \\ 0 & 0.5 & 0 & 0.5 \end{bmatrix}$$

Iteration 3 (solution found):

$$\mathbf{L}^{(3)} = \begin{bmatrix} 1 & 0 & 0 & 0 \\ 0 & 0 & 1 & 0 \\ 0 & 1 & 0 & 0 \\ 0 & 0 & 0 & 0 \end{bmatrix}, \mathbf{P}^{(3)} = \begin{bmatrix} 1 & 0 & 0 & 0 \\ 0 & 0 & 1 & 0 \\ 0 & 1 & 0 & 0 \\ 0 & 0 & 0 & 1 \end{bmatrix}$$

Next, we provide a theoretical time complexity analysis of the proposed PTGI algorithm. In each iteration of gradient descent, the most complex computation is the multiply of three matrices, i.e., $\mathbf{PBP}^T$ (line 6). The time complexity of this step is $O(n^3)$. There are at most $n+1$ iterations until a valid graph isomorphism is found or finally not found. Therefore, the worst-case and average-case time complexity of PTGI is $O(n^4)$. In the best case when the graphs prossess no symmetries, a valid graph isomorphism might be found in the first iteration. Therefore, the best-case time complexity of PTGI is $O(n^3)$. PTGI has a space complexity of $O(n^2)$ to store $n \times n$ matrices in computer memory.

It is worth to note that PTGI is an approximate graph isomorphism algorithm because it does not guarantee that an isomorphism between two isomorphic graphs can be found. It only guarantees that two non-isomorphic graphs will not be identified as isomorphic because PTGI returns true if and only if a valid isomorphism is found. Therefore, PTGI can only yield false negatives but no false positives.

## 5 EXPERIMENTAL RESULTS

### 5.1 DATASETS

We evaluate our proposed PTGI algorithm on both synthesized and real-world graphs. The set of synthesized graphs consists of random Bernoulli graphs. Each pair of vertices in a random Bernoulli graph has a 50% probability to be connected. Note that in practice we found that this vertex connection probability has little impact on the performance of our PTGI algorithm. Therefore, we do not report evaluation results given other vertex connection probabilities.

We also evaluate our algorithm on several real-world graph collections from The Network Repository Rossi & Ahmed (2015), which is a collection of network datasets covering a wide range of domains, including social networks, biological networks, transportation networks, and more. Many datasets in the repository contain graphs with hundreds to thousands of nodes.

- CHEMINFORMATICS: A collection of 646 biological molecules graphs.

- DIMACS: A collection of 78 graphs created by the Center for Discrete Mathematics and Theoretical Computer Science (DIMACS) used for benchmarking graph algorithms.

- BIOLOGICAL : A collection of 37 biological networks.

Statistics of these graphs are shown in Table 1.

Table 1: Dataset Characteristics

| Dataset | # Graphs | # Nodes | # Edges |
|---|---|---|---|
| CHEMINFORMATICS | 646 | 4-60 | 18-240 |
| DIMACS | 78 | 0.1-4K | 2K-4M |
| BIOLOGICAL | 37 | 1-43K | 1K-14M |

For each graph, we generate 50 isomorphic graphs by randomly permutate its vertices. Then, the adjacency matrix of each original graph and one of its isomorphic graph is provided to the PTGI algorithm as input. PTGI then tries to identify whether they are isomorphic and the overall accuracy and average runing time for each class of graph are reported. Note that PTGI cannot identify non-isomorphic graphs as isomorphic because it returns a true value only if it finds an exact isomorphism mapping. Therefore, it never produces false positive predictions. Consequently, we don't need to feed in non-isomorphic graph pairs for evaluation.

### 5.2 ALGORITHM IMPLEMENTATION AND PEER METHODS

We configure the parameters of the PTGI algorithm in Alg. 1 as follows. Maximum gradient descent steps $T = 100$, learning rate $\eta = 0.1$, self-supervision weight $\alpha = 1$. We implement the PTGI

algorithm using Python and TensorFlow[1]. The code is publicly available on Github.[2] The PTGI algorithm is run on a 2023 MacBook with M2 chip for running time evaluation.

We compare some of the state-of-art optimization-based graph isomorphism algorithms based on convex relaxations.

- UMEY (Umeyama's algorithm) Umeyama (1988): An eigendecomposition weighted graph matching algorithm.
- PATH Zaslavskiy et al. (2008): A convex-concave programming approach for the graph matching problem.
- QAP Vogelstein et al. (2011): An approximate graph matching algorithm via fast quadratic programming. It is equivalent to a variant of our proposed PTGI algorithm without self-supervision.

Note that the above-mentioned graph matching algorithms are also suitable for graph isomorphism problem.

## 5.3 RESULTS

The accuracy of graph isomorphism (GI) identification for synthesized and real-world graphs are presented in Tables 2 and 3, respectively.

Table 2: GI Identification Accuracy on Synthesized Graphs

| # Nodes = 100 | |
| --- | --- |
| Algorithm | Accuracy |
| UMEY | 90% |
| PATH | 92% |
| QAP | 95% |
| PTGI | 100% |
| **# Nodes = 1K** | |
| Algorithm | Accuracy |
| UMEY | 88% |
| PATH | 90% |
| QAP | 93% |
| PTGI | 100% |
| **# Nodes = 10K** | |
| Algorithm | Accuracy |
| UMEY | 87% |
| PATH | 89% |
| QAP | 93% |
| PTGI | 100% |

For synthesized graphs, the PTGI algorithm achieves perfect accuracy (100%) across all three graph sizes (100, 1K, and 10K nodes), indicating its robustness and effectiveness in identifying isomorphic graphs. The UMEY, PATH, and QAP algorithms also exhibit high accuracy, although slightly lower than PTGI, especially for larger graphs. In contrast, the accuracy of GI identification for real-world graphs varies across different domains. In the CHEMINFORMATICS domain, all algorithms achieve relatively high accuracy, with PTGI again demonstrating the highest accuracy among them. The DIMACS domain shows similar patterns, with slightly lower accuracies across the board compared to CHEMINFORMATICS. The BIOLOGICAL domain exhibits the highest accuracies

---

[1]https://www.tensorflow.org

[2]https://github.com/yangliuiuk/ML/blob/main/gi.py

Table 3: GI Identification Accuracy on Real-World Graphs

| CHEMINFORMATICS | |
| --- | --- |
| Algorithm | Accuracy |
| UMEY | 76% |
| PATH | 86% |
| QAP | 85% |
| PTGI | 90% |
| **DIMACS** | |
| Algorithm | Accuracy |
| UMEY | 75% |
| PATH | 83% |
| QAP | 87% |
| PTGI | 90% |
| **BIOLOGICAL** | |
| Algorithm | Accuracy |
| UMEY | 78% |
| PATH | 86% |
| QAP | 85% |
| PTGI | 91% |

overall, with all algorithms achieving accuracies above 85%. Once again, PTGI consistently outperforms the other algorithms in terms of accuracy across all domains. Overall, these results suggest that the PTGI algorithm is particularly effective for both synthesized and real-world graphs, consistently achieving high accuracy in GI identification tasks.

Additionally, we provide the running time of the proposed PTGI algorithm on synthesized graphs in Fig. 2. It's noteworthy that the running time primarily correlates with the number of nodes. Consequently, the running time on real-world graphs exhibits a similar pattern.

Figure 2: Running time of PTGI on Synthesized Graphs.

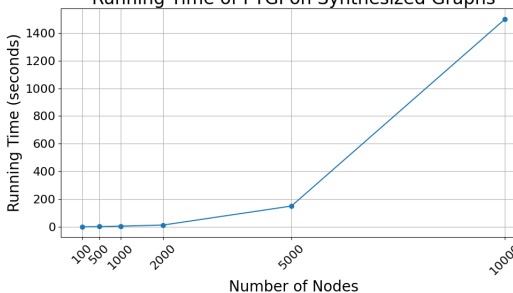

From the figure we can find that the running time of PTGI exhibits a polynomial increase pattern. PTGI performs quite efficiently on small to medium-sized graphs. For instance it runs less than 5 seconds for graphs with 1000 nodes, which is significantly faster than state-of-the-art peer methods (e.g. 300 seconds by QAP). In addition, PTGI still scale up to larger graphs with five to ten thousands nodes.

Overall, these results highlight the practical utility and efficiency of the PTGI algorithm in graph isomorphism identification, making it a promising tool for various applications across different domains.

## 6 CONCLUSION

In this paper, we introduced the Polynomial Time Graph Isomorphism (PTGI) algorithm, an optimization-based approach leveraging self-supervision techniques to efficiently tackle the graph isomorphism problem. PTGI aims to escape local optima caused by graph symmetries and provides high accuracy in identifying isomorphic graphs in polynomial time. Experimental results on both synthesized and real-world graph datasets demonstrated the effectiveness and efficiency of PTGI compared to state-of-the-art peer methods. Moreover, the running time analysis revealed that PTGI exhibits a polynomial increase in running time, running efficiently on small to medium-sized graphs and scaling well to larger graphs with thousands of nodes. Overall, the results suggest that PTGI is a promising tool for graph isomorphism identification tasks, offering high accuracy and efficiency across different graph types and sizes.

## 7 FUTURE WORKS

While the Polynomial Time Graph Isomorphism (PTGI) algorithm presented in this paper demonstrates promising performance in terms of accuracy and efficiency, there are several avenues for future research and improvement:

- **Scaling to Larger Graphs**: Although PTGI shows efficient performance on graphs with up to ten thousand nodes, further optimization is needed to handle even larger graphs efficiently. Exploring parallel processing techniques or distributed computing frameworks could help improve scalability.

- **Extension to Weighted Graphs**: The current version of PTGI is designed for unweighted graphs. Extending the algorithm to handle weighted graphs would broaden its applicability to a wider range of real-world scenarios.

- **Exploring Different Self-Supervision Techniques**: While self-supervision has proven effective in escaping local optima, exploring alternative self-supervision techniques or combinations thereof could further enhance the algorithm's performance.

- **Integration with Deep Learning Approaches**: Investigating the integration of deep learning techniques, such as graph neural networks, into the PTGI framework could potentially improve its ability to capture complex graph structures and enhance its performance on challenging graph isomorphism tasks.

- **Real-World Applications**: Conducting extensive evaluations of PTGI on real-world applications, such as molecular structure analysis, social network analysis, and bioinformatics, would provide valuable insights into its practical utility and effectiveness in real-world scenarios.

Exploring these directions could further advance the field of graph isomorphism and contribute to the development of more efficient and accurate graph analysis techniques.

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
