# OpenReview forum: "A Polynomial Time Graph Isomorphism Algorithm via Self-Supervised Gradient Descent"
_ICLR.cc/2025/Conference — Submitted to ICLR 2025_

### Official Review · Reviewer_Jvb6 · 2024-10-28

**Soundness:** 3
**Presentation:** 3
**Contribution:** 3
**Rating:** 6
**Confidence:** 4

**Summary:**

In this paper, a novel approximate algorithm for the graph isomorphism problem is presented. The main idea is to use a self- supervised signal that incrementally builds on a graph isomorphism by adding a vertex correspondence in each iteration. The algorithm is suboptimal as it does not guarantee that an isomorphism can be found between two isomorphic graphs.

**Strengths:**

-Nice algorithm that improves the computation of GI w.r.t. both accuracy and computation time

-Convincing experimental results

-Good literature review although some more papers for the PR community could be included (e.g. Optimal quadratic-time isomorphism of ordered graph or A decision tree approach to graph and subgraph isomorphism detection)

**Weaknesses:**

-I feel that subgraph isomorphism (or in general error-tolerant graph matching like graph kernels or other graph matching methods) have greater practical relevance. In other words, I doubt the practical relevance of the GI problem even though it is -- without a doubt -- a very important scientific topic.

- This point of criticism goes in the same direction as the point mentioned above. I wonder to what extent the GI problem is still relevant in times of GNNs?

- A particular weakness of the method is that it can only be applied to unlabeled graphs (both nodes and edges). I also wonder whether or not the labeling can be helpful for your algorithm (especially for the real world graphs)?

- In the introduction I somehow miss the most recent applications of graphs (and maybe you can mention some real world application of GI as well)

- Personally, I do not like to see sentences like "demonstrates high accuracy... in polynomial time..." In the introduction. In the abstract such an anticipation of the conclusions of the results may be appropriate, in the introduction rather not.
I would expect in the introduction to have some intuitive introduction, what is novel about the proposed method. Furthermore, I would expect to find out here what exactly the contribution is (in terms of the method). Especially, you should mention earlier that your method is a variant of the Vogelstein method (as stated on page 8)

- In the introduction it is not well declared that your method is an approximate method for the GI problem (or tu put it more generally: you should better discriminate the two families of algorithms). Actually, you mention for the first time that PTGI is an approximate GI algorithm at the end of Section 4, which is too late.

- You claim several times in your paper (in at least three places) that “we found that such type of optimization approaches often generate local optima...” Without empirical results/evidence, I find this somewhat tricky.

- The running time chart (Fig. 2) is in my opinion not well explained and not thorough enough. You should include also plots of the reference systems.

- Since th ePTGI does not guarantee the ability to find an isomorphism between isomorphic graphs, this introduces an element of uncertainty in cases where the algorithm does not find an isomorphism.

- Being a self-supervisor, the effectiveness of this technique depends heavily on the quality of the self-generated labels. This may limit performance in situations with more complex graphs.

- Despite its efficiency compared to other algorithms, the complexity remains prohibitive in many contexts on the practical side.

- The algorithm is designed for unweighted and unlabelled graphs, limiting its applicability

**Questions:**

- I would name the types of special graphs in paragraph 2 of page 1 (planar graphs, ordered graphs, etc.)

- I wonder to what extent the GI problem is still relevant in times of GNNs?

- I have the feeling that the paragraph below Table 1 (starting with For each graph..." Includes several redundancies with texts above.

- Is there a way to extend PTGI to weighted or labelled graphs? This would greatly extend the applicability of the algorithm.

- Is self-supervision really effective in highly symmetrical graphs?

---

### Official Review · Reviewer_tJaw · 2024-11-04

**Soundness:** 2
**Presentation:** 2
**Contribution:** 1
**Rating:** 1
**Confidence:** 5

**Summary:**

This paper presents the Polynomial Time Graph Isomorphism (PTGI) algorithm, an optimization-based approach designed to address the interesting and challenging problem of isomorphism. The authors claim to present a practical GI algorithm. I recommend rejection of this submission. Kindly refer to the weakness and Questions section for detailed technical comments.

**Strengths:**

Addressed an important theoretical problem

**Weaknesses:**

1) The authors present an approximate and not exact isomorphism algorithm, which may be more suited for graph-matching problems. I think certain cases might require an exact algorithm and not an approximate one. While computational complexity is important for isomorphism, the accuracy or correctness of the proposed algorithm should also be considered.
2) The paper lacks major theoretical contributions.
3) Incomplete literature review: The authors should discuss the following GCN or GNN papers on isomorphism:
- Z Chen, S Villar, L Chen, J Bruna, On the equivalence between graph isomorphism testing and function approximation with gnns, Neurips 2019
- Keyulu Xu, Weihua Hu, Jure Leskovec, and Stefanie Jegelka. How powerful are graph neural networks? ICLR 2019
- Haggai Maron, Heli Ben-Hamu, Hadar Serviansky, and Yaron Lipman. Provably powerful graph networks, Neurips 2019
- Ryan Murphy, Balasubramaniam Srinivasan, Vinayak Rao, and Bruno Ribeiro. Relational pooling for graph representations, ICML 2019
- Charilaos I Kanatsoulis and Alejandro Ribeiro. Graph neural networks are more powerful than we think.
- Feng, J., Chen, Y., Li, F., Sarkar, A., and Zhang, M. How powerful are k-hop message passing graph neural networks.
- Huang, Y., Peng, X., Ma, J., and Zhang, M. Boosting the Cycle Counting Power of Graph Neural Networks with I^2-GNNs.
- Zhou, C., Wang, X., Zhang, M. From Relational Pooling to Subgraph GNNs: A Universal Framework for More Expressive Graph Neural Networks
4) Comparison from existing baselines: The authors should compare their method to the above (relevant) baselines/papers.
5) Comparison on well-known datasets:
- Kindly see the circular skip link CSL dataset from Murphy et al, ICML 2019
- Kindly see the BREC dataset. Refer: Wang, Y., and Zhang, M. Towards Better Evaluation of GNN Expressiveness with BREC Dataset.
6) In line 61, the authors claim “in polynomial time, without constraints on graph types or properties”. I'm afraid that's obviously not right because the isomorphism problem can not be solved in polynomial time. Do the authors have any theoretical proof for this claim? I believe the experiments are done on the family of graphs where the proposed algorithm does not fail. Hence, the authors are not discussing the graphs for which the proposed algorithm fails.
7) Nauty is widely accepted as the practical GI algorithm. Refer [https://www.sciencedirect.com/science/article/pii/S0747717113001193](https://www.sciencedirect.com/science/article/pii/S0747717113001193). The authors should compare their work to Nauty at least empirically to claim improvement on practical graph isomorphism.
8) The average run time is almost linear for GI, but the proposed algorithm has an average runtime of $O(n^4)$.
9) The worst run time of the proposed algorithm might be better than exponential, but the average run time is still worse. I feel that the authors are doing unfair comparisons.
10) Various presentation issues: The contributions on lines 64-71 can be shortened. I feel it is too verbose. Tables 2 and 3 entries can be arranged horizontally to save some space and enhance readability.

**Questions:**

1) In line 70, the authors say various graph types. What exactly are various graph types?
2) In line 117, authors say “real-world graphs are not neccessarily “friendly””. Can the authors provide any theoretical justification or concrete arguments rather than just a claim?
3) Does their algorithm work on graphs with repeating eigenvalues? If yes, what is the proof?
4) What happens if we try random initialization for graphs shown in Figure 1?
5) In lines 342-346, the authors claim that only little impact can be seen on the performance. What is the little impact? This probably implies that the data generation process did not generate challenging cases.
6) The authors claim that the computational complexity of their method is $O(n^4)$ but isn’t it $O(T \times n^4)$? What is T chosen in Algorithm 1?
7) How does your algorithm perform on strongly regular graphs or distance regular graphs?

---

### Official Review · Reviewer_DVdY · 2024-11-04

**Soundness:** 2
**Presentation:** 1
**Contribution:** 1
**Rating:** 3
**Confidence:** 4

**Summary:**

The paper introduces the Polynomial Time Graph Isomorphism (PTGI) algorithm, which aims to solve the graph isomorphism (GI) problem through an optimization-based approach. PTGI uses self-supervision mechanism along with  gradient descent, to address the challenges of local optima caused by graph symmetries. The algorithm  demonstrates high accuracy in identifying isomorphic graphs across synthesized and real-world datasets. The authors provide empirical evidence of PTGI’s efficiency and accuracy, comparing its performance favorably against state-of-the-art convex relaxation-based GI algorithms.

**Strengths:**

Effective on Specific Datasets: PTGI shows improved accuracy in benchmark datasets for synthesized and real-world graphs, particularly when compared to convex-relaxation-based GI algorithms.

**Weaknesses:**

*Insufficient Baseline Comparisons*: The paper evaluates PTGI primarily against other optimization-based GI algorithms, without considering approximate GED solvers using either neural and combinatorial approaches. Given that the algorithm incrementally constructs node alignments, it is unclear what advantages it offers over traditional branch-and-bound GED solvers.

*Missing Discussion on Relevant Literature*: The paper is missing discussion/baseline using entropic regularization with Gumbel noise, which is known to aid in symmetry breaking and could potentially address the limitations mentioned in section 3.3.

*Numerous Typos and Incomplete Proofreading*: The manuscript contains multiple typographical errors, such as in lines 104 and 138.

*Misleading Polynomial-Time Claim*: The title and naming of PTGI imply a general polynomial-time solution to the GI problem. However, PTGI is an approximate solver, which should be clearly reflected in the title and framing to avoid misinterpretation.

**Questions:**

*Algorithm (Line 13)* : How is Line 13 implemented? The algorithm appears to contain an implicit loop that could hinder scalability, and row-based argmax does not satisfy column constraints.

*Pareto Trade-off Analysis*: Have  the authors tried generating Pareto trade-off plots showing runtime vs. accuracy across methods?

---

### Official Review · Reviewer_j5kw · 2024-11-06

**Soundness:** 2
**Presentation:** 2
**Contribution:** 2
**Rating:** 6
**Confidence:** 3

**Summary:**

The authors present a novel algorithm for the graph isomorphism problem *with one-sided error*.
The algorithm relaxes the problem as a fractional node alignment problem. As a novel approach, called self-supervision, an iteratively increasing number of node alignments is fixed and added as a constraint to the loss function. The empirical results show that the method outperforms other inexact graph isomorphism algorithms.

**Strengths:**

- very strong empirical performance
- interesting optimization approach
- paper is well-structured

**Weaknesses:**

- the paper title really oversells the result
- the paper seems repetitive in several places
- runtime experiments don't include competitor methods
- the paper should be proofread for obvious typos and missing whitespace

**Questions:**

I am sorry, I have posted a review for another paper and did not notice for a long time.

# Questions
Can you please change the title? It suggests to me that you (exactly) solve the graph isomorphism problem in polynomial time.. You do not. Suggestions might be
- An inexact ...
- ... algorithm with one-sided error via ...

I think that there is beauty in short and concise papers. However, many things are said several times. I suggest to reduce the redundancy in the writeup. E.g., graph isomorphism is defined twice (Sec. 1 and 3.1). The claim 'that in practice [optimization based GI algorithms] often result in local optima when graphs possess
symmetries' can be found in (I believe) four places at lenghts. Etc.
I suggest to instead extend the discussion on your reasoning for the newly introduced self-supervision part of the algorithm (lines 225-235) beyond one example graph with four nodes.

Algorithm 1 is very helpful. However, I am wondering if it really represents what you want to do. I assume that in line 9 you should write $\Pi = Onehot(Softmax(S))$, as $P$ was not yet updated in the inner loop and you loose one epoch of optimization.

I suggest to include the number of gradient descent steps in the runtime analysis at the top of page 7. While you can technically consider it a constant, it is a user definable parameter that you don't have any influence over.

Can you include runtime results for the competing methods in Figure 2? This would allow to put the very good empirical results of the algorithm into context.

Can the algorithm be extended to handle node labeled graphs?



# Minor issues and typos
- the paper should be proofread. There are multiple typos
- the Bernoulli graphs that you generate in the experiments are generally also known as Erdos-Renyi graphs. It may be worth mentioning this alternative name
- table 1 should include statistics for the generated graph datasets
- could you please reformat tables 2 and 3? the three datasets fit in rows next to each other
- the paragraph in lines 367-373 contains particularly many typos
- whitespace around citations is often missing. Also, I recommend distinguishing between 'silent' and 'loud' citations in the style you are using. That is Myself et. al (2024) have said that ... vs. as everybody knows (Myself et al., 2024).

---

### Meta-Review · Area_Chair_sr4R · 2024-12-18

**Metareview:**

The paper introduces an optimization-based algorithm, called Polynomial Time Graph Isomorphism (PTGI), aimed at addressing the graph isomorphism (GI) problem. It relaxes GI into a fractional node alignment task and employs a self-supervision strategy, iteratively fixing an increasing number of node alignments as constraints to guide the solution away from local optima caused by graph symmetries. The authors report that PTGI improves upon existing inexact GI algorithms, achieving high accuracy on both synthetic and real-world datasets. Despite these claims, the reviewer expresses skepticism, suggesting that the paper’s contributions are insufficient and ultimately recommends rejecting the submission.

The proposed method may have some merits. However, as pointed out by the reviewers, it is a heuristic approach and would benefit from a theoretical guarantee. Additionally, the experimental results are not sufficient. Moreover, the authors did not respond to the reviewers' concerns, and these concerns remain. Therefore, I encourage the authors to revise the paper based on the reviewers' comments and resubmit it to a future venue.

**Additional Comments On Reviewer Discussion:**

The authors did not respond and there was no discussion.

---

### Decision · Program_Chairs · 2025-01-22

Reject